

# Ground-based FTIR retrievals of SF$_6$ at Réunion Island

Minqiang Zhou[1,2,3], Bavo Langerock[2], Corinne Vigouroux[2], Pucai Wang[1,3], Christian Hermans[2], Gabriele Stiller[4], Kaley A. Walker[5], Geoff Dutton[6], Emmanuel Mahieu[7], and Martine De Mazière[2]

[1]Key Laboratory of Middle Atmosphere and Global Environment Observation, Institute of Atmospheric Physics, Chinese Academy of Sciences, Beijing, China
[2]Royal Belgian Institute for Space Aeronomy (BIRA-IASB), Brussels, Belgium
[3]University of Chinese Academy of Sciences, Beijing, China
[4]Karlsruhe Institute of Technology, Karlsruhe, Germany
[5]University of Toronto,Toronto, Canada
[6]Earth System Research Laboratory, NOAA, Boulder, Colorado, USA
[7]Institut d'Astrophysique et de Géophysique, Université de Liège, Liège, Belgium

*Correspondence to:* Minqiang Zhou (minqiang.zhou@aeronomie.be)

**Abstract.** SF$_6$ total columns are successfully retrieved from FTIR measurements (Saint Denis and Maïdo) at Réunion Island (21°S, 55°E) between 2004-2016 using the SFIT4 algorithm: the retrieval strategy and the error budget are presented. The FTIR SF$_6$ retrieval has independent information in only one individual layer, covering the whole troposphere and the lower stratosphere. The trend of SF$_6$ is analysed based on the FTIR retrieved dry air column-averaged mole fractions ($X_{SF_6}$) at

Réunion Island, the in-situ measurements at America Samoa (SMO) and the collocated satellite measurements (MIPAS and ACE-FTS) in the southern tropics. The SF$_6$ annual growth rate from FTIR retrievals is 0.265±0.013 pptv/year for 2004–2016, which is slightly weaker than that from the SMO in-situ measurements (0.285±0.002 pptv/year) for the same time period. The SF$_6$ trend in the troposphere from MIPAS and ACE-FTS observations is also close to the ones from the FTIR retrievals and the SMO in-situ measurements.

## 1 Introduction

Sulfur hexafluoride (SF$_6$) is very stable in the atmosphere and is one of the well-mixed most potent greenhouse gases listed in the 1997 Kyoto protocol linked to the United Nations Framework Convention on Climate Change (UNFCCC). It has an extremely long lifetime of 850 years (Ray et al., 2017) with Global Warming Potential for a 100-years time horizon of 23700 (relative to CO$_2$) (Kovács et al., 2017). Since SF$_6$ is very stable trace gas in the atmosphere and its annual growth rate seems

relatively constant during the last two decades (Hall et al., 2011), it can be used to calculate the age of air (Patra et al., 1997; Engel, 2002; Patra et al., 2009; Stiller et al., 2012; Haenel et al., 2015).

SF$_6$ is emitted from anthropogenic sources at the Earth's surface, mainly from the chemical industry, such as production of electrical insulators and semi-conductors, and magnesium manufacturing. The mole fraction of SF$_6$ in the atmosphere keeps increasing in recent years and the globally averaged near-surface SF$_6$ volume mixing ratio (VMR) has reached up to 7.6 pptv

(parts per trillion by volume), with an annual increase of 0.3 pptv/year in 2012 (WMO, 2014). Fig. 1 shows the SF$_6$ historical global emissions in 1900-2005 (Schultz et al., 2008; Mieville et al., 2010). Emissions of SF$_6$ started in the 1940's and have





been increasing since then. Only during the 1990-2000′s the emissions almost remain constant. The most likely reason is that $SF_6$ emissions reduced in developed countries between 1995 and 1998, but then increased again after 1998 (Levin et al., 2010; Rigby et al., 2010). The $SF_6$ global total emissions in 2005 were 6.341 Gg/year (1 Gg = 1000 tons), which is about eight times larger than that in 1970 (0.789 Gg/year). Fig. 1 also shows the predictions of $SF_6$ global emissions for 2005-2100 according

to four Representative Concentration Pathways (RCP) scenarios with different radiative forcing values (2.6, 4.5, 6.0 and 8.5 W/m$^2$) in 2100 relative to pre-industrial values (Moss et al., 2010). RCP 6.0 and RCP 8.5 scenarios assume the emissions keep increasing until 2020 and 2100 respectively, while RCP 2.6 and RCP 4.5 scenarios assume that there will be a steep decrease after 2010. The predictions from these 4 scenarios are very different, so that it is very important to monitor the abundance of $SF_6$ in the atmosphere. The most recent Scientific Assessment of Ozone Depletion report (WMO, 2014) points out that the

global emissions have amounted to 8.0 Gg/year in 2012, marked by a black dot in Fig. 1.

The Advanced Global Atmospheric Gases Experiment (AGAGE) gas chromatographic-mass spectrometric (GC-MS) system has been applied to measure the $SF_6$ concentration since 1973 (Rigby et al., 2010). The Halocarbons and other Atmospheric Trace Species Group (HATS) started $SF_6$ sampling measurements at eight stations in 1995 and in-situ measurements at six fixed sites in 1998 (Hall et al., 2011). The flask and in-situ measurements show that the $SF_6$ abundance in the atmosphere

has been increasing since the 1970s (Maiss and Levin, 1994; Geller et al., 1997; Maiss and Brenninkmeijer, 1998; Moss et al., 2010). In recent decades, remote sensing techniques also contribute to monitoring $SF_6$. Rinsland et al. (1990) used the spectra observed by the Atmospheric Trace MOlecule Spectroscopy instrument (ATMOS) aboard a shuttle to retrieve $SF_6$ concentrations in the upper troposphere and lower stratosphere. In addition, space-based sensors, such as the Atmospheric Chemistry Experiment–Fourier Transform Spectrometer (ACE-FTS) (Bernath et al., 2005; Bernath, 2017) and the Michelson

Interferometer for Passive Atmospheric Sounding (MIPAS) (Stiller et al., 2008), are applied to obtain an $SF_6$ global distribution and trend. Zander et al. (1991) succeeded in monitoring the increasing total column of $SF_6$ using the ground-based Fourier transform infrared spectrometer (FTIR) at Jungfraujoch (46.55°N, 7.98°E, 3.58 km a.s.l.). Later on, Rinsland et al. (2003) and Krieg et al. (2005) obtained the total columns of $SF_6$ from the FTIR measurements at Kitt Peak (31.9°N, 111.6°W, 2.09 km a.s.l.) and Ny–Ålesund (78.91°N, 11.88°E, 0.02 km a.s.l.). They found that the mixing ratio of $SF_6$ is continuously

increasing and that the mean increases of $SF_6$ is 0.31±0.08 pptv/year at Ny–Ålesund, 0.24±0.01 pptv/year at Jungfraujoch, and 0.28±0.09 pptv/year at Kitt Peak from March 1993 to March 2002. In the latest Scientific Assessment of Ozone Depletion, the trends of $SF_6$ from in-situ measurements are consistent with the trends in the troposphere from remote sensing measurements (ACE-FTS, MIPAS and Jungfraujoch FTIR) (WMO, 2014).

In this paper, we investigate the $SF_6$ retrievals in the southern tropics based on the spectra observed by two FTIR spectrom-

eters at Réunion Island (21°S, 55°E) from 2004 to 2016. In sect. 2, $SF_6$ retrievals are carried out with the well-established SFIT4 algorithm, which is upgraded from the radiative transfer and retrieval algorithm SFIT2 (Pougatchev et al., 1995; Hase et al., 2004) and widely used in the Network for the Detection of Atmospheric Composition Change-Infrared Working Group (NDACC-IRWG) community. The FTIR $SF_6$ retrieval strategy and the error budget will be discussed in detail. In the following section, the trend of $SF_6$ is analysed based on the FTIR retrievals, the HATS SMO in-situ measurements (14°S, 170°W, 77 m

a.s.l.) and the collocated satellite measurements (MIPAS and ACE-FTS). Finally, conclusions are drawn in Sect. 4.



## 2 FTIR retrievals at Réunion Island

The Royal Belgian Institute for Space Aeronomy operates two FTIR sites at Réunion Island. One is at Saint Denis (St Denis) close to the coast (20.90° S, 55.48° E; 85 m a.s.l.) and the other one is located at the Maïdo mountain site (21.07° S, 55.38° E; 2155 m a.s.l.). At present, both sites are equipped with a Bruker 125HR spectrometer, a precise solar-tracker system and an automatic weather station. The St Denis FTIR is dedicated to the near-infrared spectral region and contributes to the Total Carbon Column Observing Network (TCCON) since September 2011, whereas the Maïdo FTIR is dedicated to the mid- to thermal infrared spectral region and has become an NDACC-IRWG instrument in March 2013. Before September 2011, a Bruker 120M instrument was operated at St Denis in the NDACC mid- to thermal infrared configuration. For detailed information about the two sites, please refer to Zhou et al. (2016) and the references therein.

The $SF_6$ retrievals use the spectra in the thermal infrared range. Therefore, we select the spectra from the Bruker 120M at St Denis (2004-2011) and from the Bruker 125HR at Maïdo (2013-2016).

### 2.1 Retrieval strategy

The ground-based FTIR, with a maximum optical path difference (MOPD) of 250 cm, measures the interferogram of the direct solar radiation and transforms it to a spectrum with a very high spectral resolution (0.0035-0.0110 $cm^{-1}$) through a fast Fourier transform (FFT) algorithm. The HgCdTe (MCT) detector collects the spectrum and one specific interference filter is used to narrow the optical band to regions of interests in order to improve the signal to noise (SNR).

In this paper, we applied the SFIT4_v9.4.4 algorithm (Pougatchev et al., 1995) to retrieve information from the spectra: it simulates the spectrum observed by the ground-based FTIR and looks for the optimum state vector (the retrieved state) to minimize the residual between the simulated and the observed spectra. Table 1 lists the retrieval window, interfering gases, spectroscopic database, a priori profile, background parameters (slope and zero level offset (zshift)) and SNR used in the SFIT4 algorithm for the $SF_6$ retrieval at St Denis and Maïdo, together with the obtained degrees of freedom of signal (DOFS).

### 2.1.1 Retrieval window

The broad unresolved Q branch of the $\nu_3$ band of $SF_6$, at 947.9 $cm^{-1}$ (Varanasi et al., 1992), is always used to retrieve $SF_6$ by remote sensing techniques. Zander et al. (1991) used 946.9-948.9 $cm^{-1}$ to do the FTIR retrieval at Jungfraujoch and Krieg et al. (2005) used 947.2-948.6 $cm^{-1}$ for Kitt Peak and Ny–Ålesund FTIR retrievals. We also use the $SF_6$ absorption line at 947.9 $cm^{-1}$ and the retrieval window 946.5-949.0 $cm^{-1}$ to perform the FTIR retrieval at Réunion Island. However, compared with the previous studies, our retrieval window contains an extra weak $H_2O$ absorption line at 946.68 $cm^{-1}$. Since there is a strong $H_2O$ absorption line at 948.26 $cm^{-1}$ and a strong $CO_2$ line at 947.74 $cm^{-1}$ (see Fig. 2), the $SF_6$ is inevitably influenced by these two species, especially from $H_2O$ due to its larger variability in the atmosphere. A better fitting of $H_2O$ is obtained by the larger retrieval window. In addition, to minimize interference from $H_2O$ and $CO_2$, their profiles are retrieved simultaneously with the $SF_6$ profile.





### 2.1.2 Instrument line shape

In order to acquire the instrument line shape (ILS) and to verify the alignment of the instrument, daily HBr cell measurements are carried out automatically at both sites. The LINEFIT14.5 program (Hase et al., 1999) is applied to obtain the modulation and phase parameters of the ILS, which are used as an input in the SFIT4 algorithm. Note that, we make a 3-order polynomial

fitting from the LINEFIT outputs, and then retrieve the polynomial parameters in SFIT4 algorithm for both modulation and phase.

### 2.1.3 Spectroscopy

The spectroscopy of $SF_6$ is taken from the Pseudo linelists (http://mark4sun.jpl.nasa.gov/pseudo.html), and the spectroscopy of the other species is obtained from the ATM16 linelists provided by G. Toon (JPL).

### 2.1.4 A priori profile

To construct the a priori profile close to the true one, we use the US standard atmosphere (1976) $SF_6$ (National Geophysical Data Center, 1992) as the shape of the a priori profile, and then scale it with one factor to make the concentration of the lowest level equal to the annual mean of SMO measurements in 2009. The $H_2O$ a priori profile is derived from the 6-hourly NCEP reanalysis data. For the a priori profiles of the other interfering species (see Table 1), the mean of the Whole Atmosphere

Community Climate Model (WACCM) version 6 monthly profiles between 1980 and 2020 are adopted.

### 2.1.5 Regularisation matrix

The a priori covariance matrix together with the measurement noise covariance matrix determine the weights of a priori knowledge and measurement information (Rodgers, 2000). The SNR are set as 180 and 400 at St Denis and Maïdo, respectively. In order to extract as much as possible information from the measurements and to avoid too many oscillations in the retrieved

$SF_6$ profiles, we use 30% and 14% as the diagonal elements (the same value for all levels) to create the regularisation matrices at St Denis and Maïdo, respectively. The correlation width is set as 10.0 km. Note that the diagonal value of the regularisation matrix is a key parameter to balance the contribution from the measurement information and the a priori information, which does not represent the real variability of $SF_6$ in the atmosphere.

### 2.1.6 Averaging kernel

Figure 3 shows the typical averaging kernel of the $SF_6$ retrieval at Maïdo. The FTIR retrieval is sensitive to the altitude range from the surface to 20 km (the whole troposphere and lower stratosphere). The mean and standard deviation of the DOFS of the $SF_6$ retrievals is $1.0 \pm 0.1$ at St Denis and $1.1 \pm 0.1$ at Maïdo, indicating that the $SF_6$ retrievals have information content in only one individual layer (mainly 0-20 km) and have no profile information. That means when we use the FTIR retrievals, only the total column should be trusted instead of the profile. In this study, the $SF_6$ retrievals at St Denis are combined with



Maïdo retrievals to extend the time coverage for the trend in Sect. 3. The DOFS at the two stations are very close, and there is no observed trend in the time series of the DOFS.

## 2.2 Error budget

Based on the optimal estimation method (Rodgers, 2000), the difference between the retrieved state vector $\hat{x}$ and the true state
vector $x_t$ could be expressed as

$$\hat{x} - x_t = (\mathbf{A} - \mathbf{I})(x_t - x_a) + \mathbf{G}_y \mathbf{K}_b (b_t - b) + \mathbf{G}_y \triangle \mathbf{f} + \mathbf{G}_y \varepsilon_y, \tag{1}$$

where $x_a$ is the a priori state vector; $\mathbf{A}$ is the averaging kernel matrix, representing the sensitivity of the retrieved state vector to the true state vector; $\mathbf{I}$ is a unit matrix; $\mathbf{G}_y$ is the contribution matrix, representing the sensitivity of the retrieval to the measurement $y$; $\mathbf{K}_b$ is the weight function, representing the sensitivity of the forward model $F(x, b)$ to the forward model parameters; $b$ is the vector of forward model parameters that are not retrieved; $b_t$ is the vector of true forward model parameters; $\triangle \mathbf{f}$ is the forward model systematic uncertainty; $\varepsilon_y$ is the measurement noise covariance matrix. Note that the state vector $x$, which is the vector of forward model parameters that are retrieved, is a higher dimensional vector which components consist of the target $SF_6$ profile components, the concentration profiles for the interfering species ($H_2O$, $CO_2$) and other retrieval parameters (slope, ILS, ...).

The error on the target $SF_6$ profile is obtained by extracting the $SF_6$ components from the vectorial equation in Eq. 1. The error on the retrieved $SF_6$ profile $(\hat{x} - x_t)_{SF_6}$ then consists of the smoothing error $(\mathbf{A} - \mathbf{I})(x_t - x_a)$, model parameter error $\mathbf{G}_y \mathbf{K}_b (b_t - b)$, forward model error $\mathbf{G}_y \triangle \mathbf{f}$ and measurement noise $\mathbf{G}_y \varepsilon_y$. As the SFIT4 algorithm is well established and only the physics of the absorption is included in the transmission of radiation, we assume that the forward model error can be ignored.

For the smoothing error, except for the uncertainty from $SF_6$, it also includes the uncertainties from the $H_2O$ profile, the $CO_2$ profile, the $C_2H_4$ and $O_3$ scaling factors and some other parameters (see Table 1), which must be taken into account when estimating the error budget of the retrieved $SF_6$. Since the absorption lines of $H_2O$ and $CO_2$ are very strong in the retrieval window, we separate the *retrieval parameter error* as three components.

$$(\mathbf{A} - \mathbf{I})(x_t - x_a) = (\mathbf{A}_{SF6,SF6} - \mathbf{I})(x_{t,SF_6} - x_{a,SF_6}) + retrieval\ parameter\ error \tag{2}$$

$$
\begin{aligned}
retrieval\ parameter\ error = {} & \mathbf{A}_{SF_6,H_2O}(x_{t,H_2O} - x_{a,H_2O}) + \mathbf{A}_{SF_6,CO_2}(x_{t,CO_2} - x_{a,CO_2}) \\
& + \mathbf{A}_{SF_6,others}(x_{t,others} - x_{a,others}),
\end{aligned} \tag{3}
$$

where $\mathbf{A}_{SF_6,SF_6}$, $\mathbf{A}_{SF_6,H_2O}$, $\mathbf{A}_{SF_6,CO_2}$ and $\mathbf{A}_{SF_6,others}$ are the matrices extracted from the full averaging kernel $\mathbf{A}$ by selecting the components $\mathbf{A}_{ij}$ where the row index $i$ runs over all $SF_6$ components in the state vector $x$ and the column index $j$ runs over all $SF_6$, $H_2O$, $CO_2$ and other components in state vector $x$. $x_{t,SF_6}$ and $x_{a,SF_6}$, $x_{t,H_2O}$ and $x_{a,H_2O}$, $x_{t,CO_2}$ and $x_{a,CO_2}$, $x_{t,others}$ and $x_{a,others}$ are the true and a priori values of $SF_6$, $H_2O$, $CO_2$ and other retrieval parameters, respectively.





Systematic and random components are considered to characterize the uncertainty of each parameter. For the smoothing error $(\mathbf{A}_{SF6,SF6} - \mathbf{I})(\boldsymbol{x}_{t,SF_6} - \boldsymbol{x}_{a,SF_6})$, we assume that the systematic uncertainty of $\boldsymbol{\varepsilon}(\boldsymbol{x}_{t,SF_6} - \boldsymbol{x}_{a,SF_6})$ is 5% relative to the a priori profile ($\sigma_{SF6,ai} = 0.05 x_{ai}$). Then, the diagonal and off-diagonal values of the systematic part of $\boldsymbol{\varepsilon}(\boldsymbol{x}_{t,SF_6} - \boldsymbol{x}_{a,SF_6})(\boldsymbol{x}_{t,SF_6} - \boldsymbol{x}_{a,SF_6})^T$ are calculated as ($\sigma^2_{SF6,ai}$) and $\sigma_{SF6,ai}\sigma_{SF6,aj}$, respectively (von Clarmann, 2014). The random part of $\boldsymbol{\varepsilon}(\boldsymbol{x}_{t,SF_6} - \boldsymbol{x}_{a,SF_6})(\boldsymbol{x}_{t,SF_6} - \boldsymbol{x}_{a,SF_6})^T$ is constructed same as the regularisation matrix but the diagonal elements are set as 30% for both St Denis and Maïdo. For the measurement error $\mathbf{G}_y\boldsymbol{\varepsilon}$, there is no systematic uncertainty and the random uncertainty is derived from SNR.

For the *retrieval parameter error*, we mainly focus on the influence from $H_2O$ and $CO_2$. The systematic and random uncertainties of $H_2O$ profile are derived from the bias and the standard deviation of the differences between the NCEP profiles and the balloon measurements. In general, the systematic uncertainty is about 5% and random uncertainty is about 10% from surface to 10 km. The $CO_2$ systematic uncertainty is assumed to be 5% of the average of the WACCM monthly profiles, and the random uncertainty is the standard deviation of the WACCM monthly profiles from 1980 to 2020.

For the model parameter error $\mathbf{G}_y\mathbf{K}_b(\boldsymbol{b}_t - \boldsymbol{b})$, we only show the significant parameters here, i.e. temperature, spectroscopy, solar zenith angle (SZA), ILS and zshift. The systematic and random uncertainties of the temperature profile are derived from the mean and the standard deviation of the differences between the NCEP profiles and the balloon measurements at Réunion Island in 2011. In general, the systematic bias is about 5 K below 10 km, 3 K between 10 km and 15 km and 1 K above 15 km. The standard deviation is about 2-4 K in troposphere and 5-10 K above tropopause height. The $SF_6$ spectroscopy uncertainty is from the Pseudo database, 2% for the systematic part and zero for the random part. 0.1% and 0.2% are adopted for the systematic and random uncertainties of SZA, which are provided by the Pysolar package (one Python code to calculate the solar position http://pysolar.org/). 5% and 1% are adopted for both systematic and random uncertainties of the ILS parameters and zshift, respectively.

Table 2 lists the $SF_6$ FTIR retrieval systematic and random uncertainties (%) at St Denis and Maïdo. The "retrieval parameters" in the Table 2 represents the "others" in Eq.3. The smoothing error, measurement error, $H_2O$ interfering and temperature error at St Denis are much larger than those at Maïdo. In total, the retrieval systematic/random uncertainties (relative to the retrieved $SF_6$ total column) are 4.6%/14.0% at St Denis and 3.7%/6.7% at Maïdo, respectively.

## 3  $SF_6$ trend analysis

### 3.1  Data sets

#### 3.1.1  SMO in-situ measurements

Since 1998, a four channel gas chromatograph (CATS) system has been measuring the surface $SF_6$ at the SMO site. Due to the high accuracy and precision, the CATS $SF_6$ daily data from the NOAA/ESRL halocarbons in situ measurement program is considered to be a reference for comparison with the FTIR retrievals. Note that these are daily medians data instead of daily means, in order to filter the higher outliers from local pollutions. As there is an improvement of the instrument in June 2000,





the standard deviation of one-day's measurements decreased from 0.2-0.4 pptv to 0.02-0.04 pptv after the change (Hall et al., 2011).

### 3.1.2   MIPAS

MIPAS derived the global distributions of profiles of $SF_6$ from limb observations between 2002 and 2012. MIPAS observed
spectra in full spectral resolution (FR) mode (spectral resolution: 0.05 cm$^{-1}$) and reduced resolution (RR) mode (spectral resolution: 0.121 cm$^{-1}$) before and after January 2005. In this paper, we use the latest $SF_6$ product with newly calibrated level 1b spectra (Haenel et al., 2015) to compare with the FTIR retrievals and to make the $SF_6$ trend and seasonal cycle analysis. The $SF_6$ data used here are version V5h_SF6_20 for the FR data product and V5r_SF6_222 and V5r_SF6_223 for the RR period. The MIPAS retrievals cover the upper troposphere (down to cloud top, or ∼6 km in cloud free cases) and the stratosphere only
(about 55 km; see Fig. 6). Since MIPAS single $SF_6$ profiles are very noisy, we use the monthly means in the latitude band of 20-25° S.

### 3.1.3   ACE-FTS

Global distributions of $SF_6$ were also monitored by ACE-FTS occultation measurements from 2004 (Boone et al., 2013). We use the ACE-FTS level 2 version 3.5 monthly data (2004-2013) from the ACE/SCISAT database, and only the measurements
without any known issues (quality flag = 0) are selected (Sheese et al., 2015). The ACE-FTS data has been validated with MkIV balloon profiles (Velazco et al., 2011). Since ACE-FTS looks at the polar area, there are few measurements in the tropical zone. Geller et al. (1997) found that $SF_6$ is well-mixed throughout the southern hemisphere, therefore, we enlarge the latitude band for ACE-FTS measurements to 0-40° S to get a robust result. Similar to MIPAS measurements, ACE-FTS mainly collects the spectra in the upper troposphere and stratosphere (about 10-30 km; see Fig. 6).

### 3.1.4   Ground-based FTIR

As the FTIR $SF_6$ retrievals have only one-layer's information , we consider the dry air column-averaged $SF_6$ ($X_{SF_6}$) obtained by dividing the $SF_6$ total column by the dry air total column, for quantitative comparisons with the other data sets. Note that the $SF_6$ concentration is almost constant in the troposphere, but much lower in the stratosphere. Such a kind of profile will lead to a systematic bias if we combine the $X_{SF_6}$ in 0-100 km (above St Denis) and $X_{SF_6}$ in 2.155-100 km (above Maido) directly. To
avoid such systematic bias, we keep the $X_{SF_6}$ at St Denis unchanged and apply a scaling factor of 1.01 to the $X_{SF_6}$ at Maïdo, which is the ratio of $X_{SF_6}$ in 0-100 km to $X_{SF_6}$ in 2.155-100 km based on the FTIR $SF_6$ a priori profile but scaled with the annual mean of SMO in-situ measurements in 2014.

Figure 4 shows $SF_6$ time series of SMO in-situ, MIPAS and ACE-FTS measurements and FTIR retrievals at St Denis and Maïdo. For MIPAS, ACE-FTS and FTIR data, the errorbar is the standard deviation of all the measurements in one month.
Since the FTIR retrieval has the largest sensitivity in the vertical range of 5-15 km (see Fig. 3), we select the 11 km of MIPAS





and 12.5 km of ACE-FTS here. In general, $SF_6$ from these data sets are in good agreement, as the difference between each two measurements is within the their uncertainties.

## 3.2 Methodology

A regression model is applied to derive the $SF_6$ linear long-term trend based on the measurements of FTIR daily means, SMO daily medians and satellites (MIPAS and ACE-FTS) monthly means.

$$\boldsymbol{Y}(\boldsymbol{t}) = A_0 + A_1 \cdot \boldsymbol{t} + \sum_{k=1}^{3} (A_{2k} \cos(2k\pi\boldsymbol{t}) + A_{2k+1}\sin(2k\pi\boldsymbol{t})) + \boldsymbol{\varepsilon}(\boldsymbol{t}), \tag{4}$$

where $\boldsymbol{Y}(\boldsymbol{t})$ is measurements with the $\boldsymbol{t}$ in fraction of year; $A_0$ is the intercept; $A_1$ is the annual growth; $A_2$ to $A_7$ are the periodic variations, mainly representing the seasonal cycle; $\boldsymbol{\varepsilon}(\boldsymbol{t})$ is the residual between the measurements and the fitting model. To estimate the trend error $\sigma_c$, the auto-correlation of the residual should be taken into account (Santer et al., 2000).

$$\sigma_c = \sigma_d \frac{(n-2)}{[n(1-r)/(1+r)-2]}, \tag{5}$$

where $\sigma_d$ is the regression error; $n$ is the number of measurements; $r$ is the lag-1 (one month) auto-correlation coefficient of the regression residuals.

## 3.3 Annual change

Figure 5 shows the $SF_6$ trends from the SMO in-situ measurements, the ground-based FTIR retrievals, the MIPAS measurements in the latitude band of 20-25° S for different altitudes (9-52 km), and the ACE-FTS measurements in the latitude band of 0-40° S for altitude range of 10.5-32.5 km. The vertical sensitivity of the FTIR retrieval is between surface and 20 km (see Fig. 3). For MIPAS and ACE-FTS measurements, Fig. 5 also shows the number of monthly means used for the trend analysis at each altitude (dotted lines). The annual growth of FTIR measurements is 0.265±0.013 pptv/year from 2004 to 2016, which is slightly weaker than the trend of the SMO in-situ measurements (0.285±0.002 pptv/year) for the same time period. Waugh et al. (2013) pointed out that the age of near-surface $SF_6$ at SMO (14°S) is about 0.4 years greater than that at Réunion Island (14°S). In addition, the global surface in-situ measurement network (https://www.esrl.noaa.gov/gmd/hats/combined/SF6.html) shows that the growth rate of $SF_6$ is slightly increasing with time. Therefore, it is acceptable that the trends from FTIR measurements at Réunion Island is slightly weaker than that from the SMO in-situ measurements.

The trend uncertainty from MIPAS data is less than the ACE-FTS data and the FTIR retrievals because MIPAS has much more data points. The profile of $SF_6$ trend shows a peak in 11-13 km altitude from the MIPAS measurements, and a peak in 11.5-16.5 km from the ACE-FTS measurements. As the $SF_6$ emissions are all at Earth's surface and there is almost no removal mechanism in the troposphere and stratosphere (Kovács et al., 2017), the $SF_6$ concentration should be well-mixed in the troposphere (the tropopause height above Réunion Island is about 16.5 km) and decreasing above tropopause, which was confirmed by the airborne in-situ measurements (Patra et al., 1997). Fig. 6 shows the $SF_6$ monthly means and the number of measurements in each month from MIPAS and ACE-FTS. The numbers of good quality measurements at 9 km for MIPAS




and 10.5 km for ACE-FTS are considerably reduced because a large number of measurements are contaminated by clouds. As a consequence, the trends at these altitudes from MIPAS and ACE-FTS are derived from a small number of measurements, leading to larger uncertainties. For example, in October 2004, there are only 3 ACE-FTS measurements within the latitude band range 0-40° S, and the $SF_6$ monthly mean at 10.5 km is 7.57 pptv, which is very large compared with the monthly means

nearby in time (the $SF_6$ monthly means at 10.5 km in November and December are 4.92 and 5.80 pptv).

In general, the $SF_6$ trend from the SMO in-situ measurements at surface or from the FTIR retrievals is close to the trends at the troposphere from the MIPAS and ACE-FTS measurements. In the stratosphere, satellite measurements (both MIPAS and ACE-FTS) show that the $SF_6$ trend decreases with increasing altitude. The change of the $SF_6$ trends in the stratosphere could be applied to estimate how long it takes for the well-mixed air mass to transport from surface to the high altitude in a large

scale (Waugh, 2002; Stiller et al., 2012).

## 4    Conclusions

The $SF_6$ total columns are retrieved with SFIT4 algorithm from two FTIRs at Réunion Island (21°S, 55°E) in 2004-2016. The FTIR $SF_6$ retrieval is sensitive to the whole troposphere and lower stratosphere, but has only one degree of freedom. We use the retrieval window (946.5-949.0 cm$^{-1}$) to do the $SF_6$ retrieval at St Denis and Maïdo, with the broad unresolved Q branch of

the $\nu_3$ band of $SF_6$, at 947.9 cm$^{-1}$. Nearby are a strong $H_2O$ absorption line at 948.26 cm$^{-1}$, a weak $H_2O$ absorption line at 946.68 cm$^{-1}$ and a strong $CO_2$ line at 947.74 cm$^{-1}$. The $SF_6$ retrieval product is influenced by these two species, especially by $H_2O$ due to its larger variability in the atmosphere. The retrieval window in this study is wider than the previous ones (Zander et al., 1991; Krieg et al., 2005) because for the humid sites, such as St Denis, a better fitting is obtained with the larger window.

To estimate the $SF_6$ retrieval error, four components (the smoothing error, forward model parameter error, measurement

error and other retrieval parameter errors) have been discussed in detail. In total, the systematic/random uncertainties of the FTIR retrieved $SF_6$ columns are 4.6%/14.0% at St Denis and 3.7%/6.7% at Maïdo. Both systematic/random uncertainties at St Denis are larger than those at Maïdo, because of the lower SNR and the higher water vapour abundance at St Denis.

The trend of $X_{SF_6}$ derived from FTIR measurements is 0.265±0.013 pptv/year for 2004-2016, which is slightly weaker than the trend from the SMO in-situ measurements (0.285±0.002 pptv/year) for the same time period. The $SF_6$ trends at 9 km from

MIPAS measurements and 10.5 km from ACE-FTS measurements are rather uncertain due to scarceness of data, because the MIPAS and ACE-FTS measurements are contaminated by cumulus clouds at low altitudes and these values are not included for the trend calculation. The $SF_6$ trends in the troposphere from both MIPAS and ACE-FTS measurements are close to the trends from FTIR retrievals and SMO in-situ measurements; the $SF_6$ trends from MIPAS and ACE-FTS above the tropopause height decrease with increasing altitude.





## 5 Data availability

The FTIR $SF_6$ retrievals at Réunion Island (St Denis and Maïdo) is not public available yet. To obtain access to site data, please contact the author or the BIRA-IASB FTIR group. The MIPAS $SF_6$ data is provided by the MIPAS satellite group at KIT/IMK, please contact Gabriele Stiller (gabriele.stiller@kit.edu). The ACE-FTS data used in this study are available from

5   http://ace.uwaterloo.ca/data/ (registration required). SMO in-situ $SF_6$ measurements are public available ftp://ftp.cmdl.noaa. gov/hats/sf6/insituGCs/CATS/daily/.

*Acknowledgements.* The authors thank the National Basic Research Program of China (2013CB955801), the National Natural Science Foundation of China (41575034), the Belgian Science Policy for financial support through the supplementary researchers programme and the AGACC projects (SD/AT/01A) and (SD/CS/07A) in the Science for Sustainable Development programme. They wish to thank the

10   Université de la Réunion, in particular Jean-Marc Metzger (UMS3365 of the OSU Réunion) as well as the French regional and national (INSU, CNRS) organizations, for supporting the NDACC operations in Reunion Island. We also want to thank Geoff Toon (JPL) for providing the spectroscopy. The Atmospheric Chemistry Experiment (ACE), also known as SCISAT, is a Canadian-led mission mainly supported by the Canadian Space Agency and the Natural Sciences and Engineering Research Council of Canada. MIPAS $SF_6$ data were derived within research projects funded by the "CAWSES" priority programme of the German Research Foundation (DFG) (project STI 210/5-3) and

15   the"ROMIC" programme of the German Federal Ministry of Education and Research (BMBF) (project 01LG1221B).



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



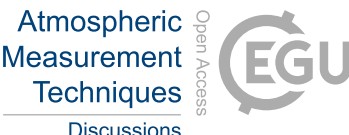

**Table 1.** The retrieval window, interfering gases, spectroscopic database, a priori profile, background parameters (slope and zshift) and SNR used in SFIT4 algorithm for FTIR $SF_6$ retrieval at St Denis and Maïdo, together with the achieved DOFS (mean and the standard deviation) of the retrievals.

| Target gas | $SF_6$ |
| --- | --- |
| Window ($cm^{-1}$) | 946.5–949.0 |
| Profile retrieval | $SF_6$, $H_2O$, $CO_2$ |
| Column retrieval | $C_2H_4$, $O_3$ |
| Spectroscopy | Pseudo, ATM16 |
| A priori profile | US standard but scaled to SMO measurements |
| ILS | LINEFIT14.5 |
| Background (St Denis/Maïdo) | slope, zshift/slope |
| SNR (St Denis/Maïdo) | 180/400 |
| DOFS (St Denis/Maïdo) | $1.0 \pm 0.1/1.1 \pm 0.1$ |



**Table 2.** The systematic and random uncertainties for the FTIR retrieved total column (%) at St Denis and Maïdo. $\sigma_b$ are the relative systematic (random) uncertainties of the non-retrieved parameters (%). The "retrieval parameters" represents the "others" in Eq.3. When a relative uncertainty is smaller than 0.1 %, it is considered negligible and represented as "–".

| Error | $\sigma_b$ | St Denis Systematic | St Denis Random | Maïdo Systematic | Maïdo Random |
|---|---|---|---|---|---|
| Smoothing | | 0.1 | 6.3 | 0.1 | 3.0 |
| Measurement | | – | 10.6 | – | 4.8 |
| Retrieval parameters | | 0.2 | – | 0.1 | 0.1 |
| $H_2O$ interfering | | 0.4 | 6.1 | 1.0 | 3.3 |
| $CO_2$ interfering | | – | 0.2 | – | 0.1 |
| Temperature | | 4.1 | 2.0 | 2.5 | 1.0 |
| $SF_6$ spectroscopy | 2(0) | 2.2 | – | 2.2 | – |
| SZA | 0.1(0.2) | 0.2 | 0.4 | 0.3 | 0.6 |
| ILS | 5(5) | 0.2 | 0.2 | 0.2 | 0.2 |
| zshift | 1(1) | 0.2 | 0.2 | – | – |
| Total | | 4.6 | 14.0 | 3.7 | 6.7 |



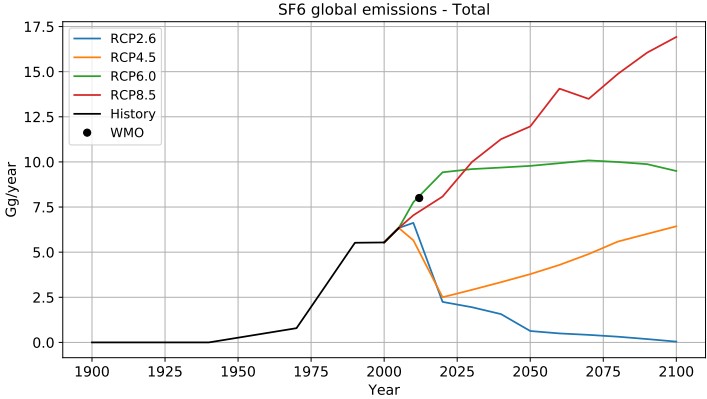

**Figure 1.** Time series of historical and projected global SF$_6$ emissions. Historical data cover 1900–2005 (black), and projections for the 2005–2100 time period correspond to four 4 RCP scenarios with 2.6, 4.5, 6.0 and 8.5 W/m$^2$ radiative forcing in 2100 relative to pre-industrial values (Moss et al., 2010). The black dot is the annual growth of SF$_6$ in 2012 according to the WMO report (WMO, 2014).

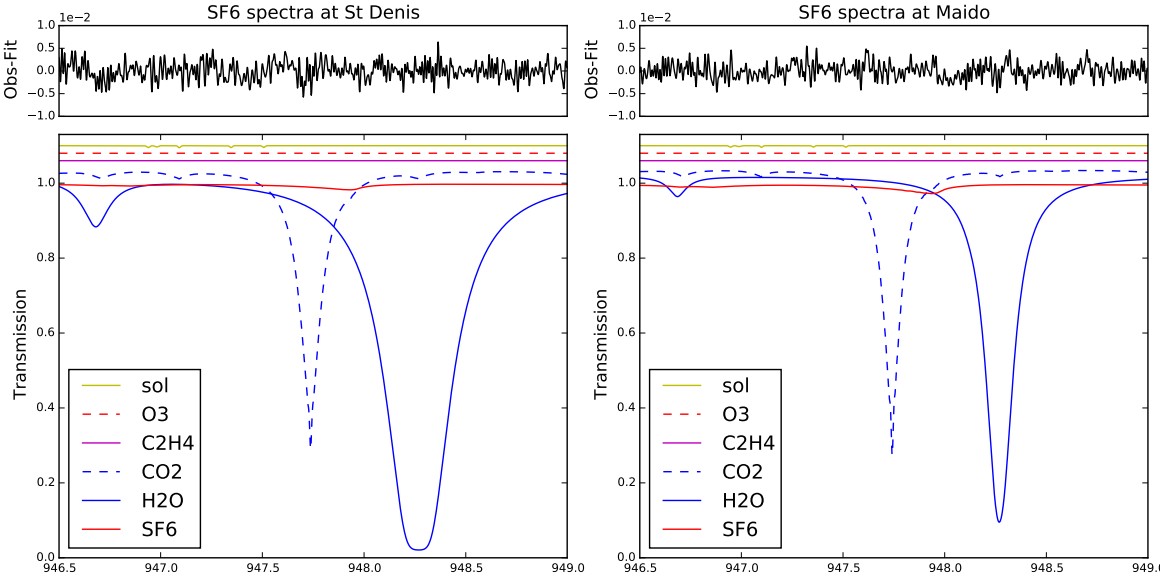

**Figure 2.** The typical spectrum of SF$_6$ retrieval microwindow (946.5-949.0 cm$^{-1}$) at St Denis (left) and Maïdo (right). The top panels show the transmittance residual (observed-calculated), and the bottom panels list the absorption contribution from each species. To clarify the absorption lines, the transmittance is shifted by 0.02 for each species and the solar (sol) line list.





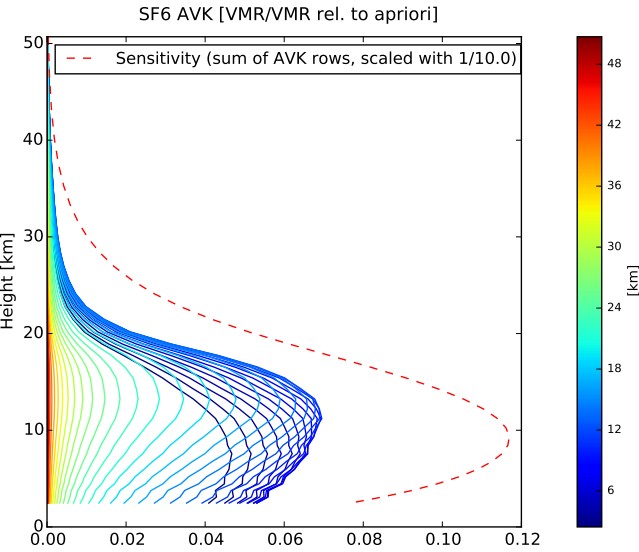

**Figure 3.** The typical averaging kernel of $SF_6$ retrieval at Maïdo. The solid lines represent the sensitivities at specific altitudes. The red dashed line is the sum of the row of averaging kernel scaled by 0.1, indicating the vertical sensitivity.

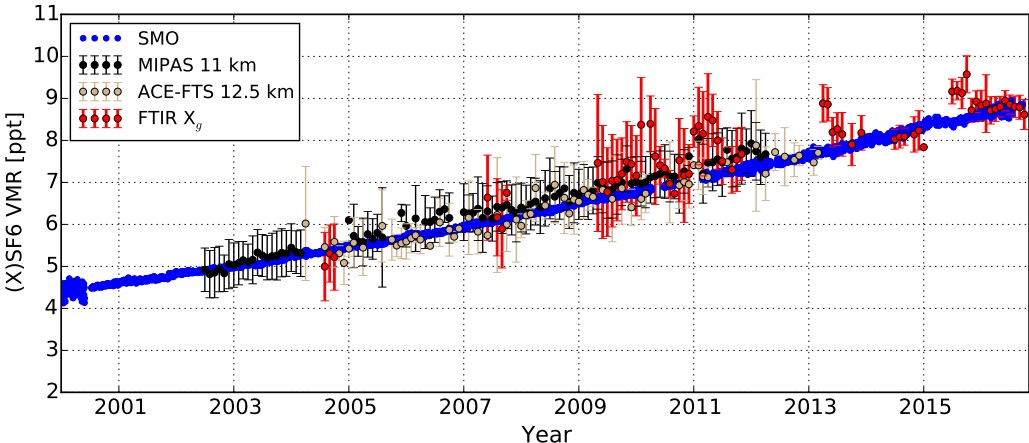

**Figure 4.** Time series of SMO in-situ $SF_6$ daily median (blue), MIPAS $SF_6$ monthly mean (20-25° S) at 11 km (black), ACE-FTS $SF_6$ monthly mean (0-40° S) at 12.5 km and FTIR $X_{SF_6}$ monthly mean at St Denis and Maïdo (red). For MIPAS, ACE-FTS and ground-based FTIR measurements, the errorbar is the standard deviation within one month.





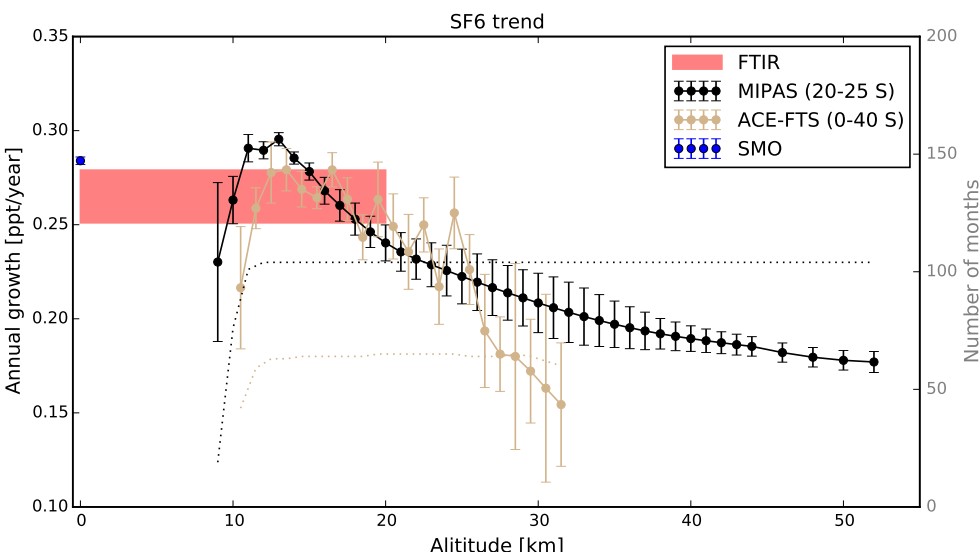

**Figure 5.** SF$_6$ annual growths from SMO in-situ measurements (2004-2016) (blue bar), ground-based FTIR measurements (2004-2016: combined St Denis and Maïdo)(pink bar), MIPAS measurements (2002-2012) in the latitude band of 20-25° S for different altitudes (9-52 km) (black solid line) and ACE-FTS measurements (2004-2013) in the latitude band of 0-40° S for altitude range of 10.5-32.5 km (brown solid line). For MIPAS and ACE-FTS measurements, the dotted line of the same colour is the number of monthly means used for trend analysis at each altitude.





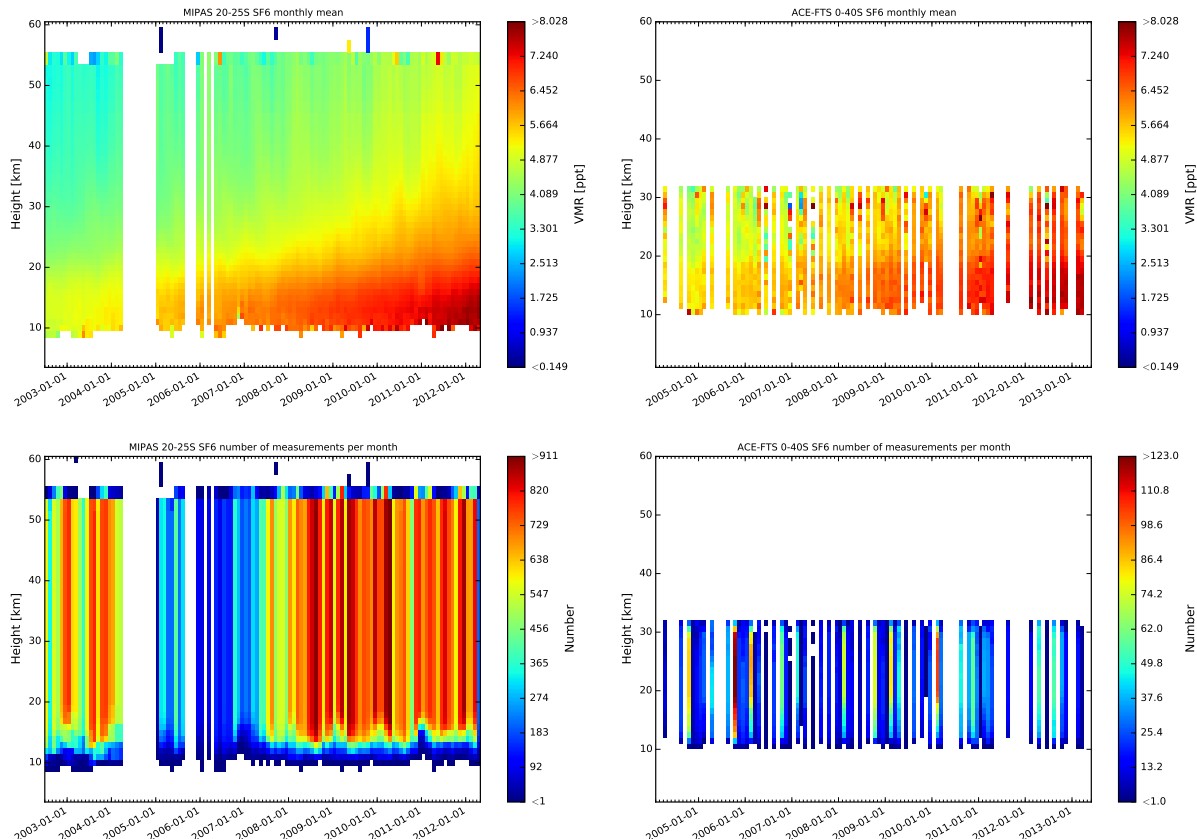

**Figure 6.** SF$_6$ monthly means of volume mixing ratios profiles (upper) and the number of measurements in each month (bottom) for MIPAS in the latitude band between 20-25° S (left) and ACE-FTS in the latitude band between 0-40° S (right).