# Peer review of "Ground-based FTIR retrievals of SF6 at Réunion Island"

_Atmospheric Measurement Techniques, 2017_

## Referee Comment (RC1) · Anonymous Referee #1 · 7 Dec 2017

The manuscript submitted by Zhou et al. presents measurements and trends of SF6 in the atmosphere using remote sensing techniques with ground-based FTIR spectrometers on Reunion Island. These are supplemented with analysis of measurements from in-situ instruments and space-borne sensors. This work is very significant and I recommend its publication after a few items below have been addressed/clarified.

General Comments

Column averaged dry mole fractions of SF6: To readers not familiar with this method, elaboration of this procedure is necessary. How are the dry air total columns measured/obtained? Since the XSF6 trend would depend on this, it would be important to establish and/or show the robustness of the dry air total column measurements.

Pseudo-lines: Please explain/justify the use of pseudo-lines, e.g. how are they derived? Do ACE and MIPAS use pseudo-lines as well?

Smoothing error: From my understanding, the smoothing error can be evaluated if there is an independent measurement of the true profile, an estimate of its variability and covariance matrix (e.g. Barrett et al., 2002 did this for Ozone, http://onlinelibrary.wiley.com/doi/10.1029/2001JD001298/abstract). Though I understand that following this would be difficult for SF6 at the site, would the authors please comment on this?

On the use of the retrieval window and improved fits: What criteria is used to determine a "better fit"? For example, in Fig. 2. (right panel), the residuals at 948.0 cm$^{-1}$ seem to indicate that something is not fitted well.

Specific comments

Instrument and ILS: A mention of the spectral resolution used in the actual retrieval and the goodness of the ILS would be helpful to readers. For example, in Section 2.1., it was mentioned that the max OPD of the instrument is 250 cm, is this the same OPD used for SF6 measurements at both sites?

Page 6 line 10 "balloon measurements" appears here for the first time. I think this should be defined earlier in the manuscript. I assume these are sondes, right?

On Table 2, Please elaborate on how the systematic and random errors under "SF6 Spectroscopy" were calculated.

Technical Comments

Page 2, line 2: I suggest to change "SF6 emissions reduced" to "SF6 emissions decreased"

Page 2, line 17: "aboard a shuttle": please specify which one, if possible.

Page 4, line 19: "In order to extract as much as possible information" -> "In order to

extract as much information as possible"

Page 4, line 28: Consider rephrasing this sentence: "That means when we use the FTIR retrievals, only the total column should be trusted instead of the profile."

---

## Referee Comment (RC2) · Anonymous Referee #2 · 30 Dec 2017

The manuscript "Ground-based FTIR retrievals of SF$_6$ at Réunion Island" by Minqiang Zhou et al. describes a 12-year time series of SF$_6$ column measurements in the troposphere and lowermost stratosphere. Given the scarce observation pool of SF$_6$ in general and its significance as a purely anthropogenic extremely long-lived greenhouse gas with a huge global warming potential, this is a very valuable data set. The authors derive an SF$_6$ trend from their own observations as well as a comparison with two satellite datasets and one set of near-surface in-situ observations in the tropics.

The manuscript is well written and describes the data sets and the retrieval parameters and error budgets for the SF$_6$ timeseries that were used in the study. To calculate trends from each dataset, a linear model with periodic (seasonal) components is applied. The resulting trend estimates are close but not identical. This is explained by different

vertical and latitudinal coverage of the used datasets.

**General comments:**

1. One criticism that I have is the misuse of tense throughout the manuscript. Practically all of the text is written in present tense. However, the convention for scientific writing is that past tense should be used for reporting the authors' observations and results while present tense is reserved for well-known facts and cited results from the scientific literature. Please refer to guidelines on the internet such as
https://www.nature.com/scitable/topicpage/effective-writing-13815989.

2. I also think that there should be a map that shows the locations of the ground based observations as well as the latitude bands covered by the satellites.

3. Given the fact that the $SF_6$ spectral lin is weak and the retrieval depends on the SMO observations for prior information, the significance of the derived trend(s) should be better scrutinized. Please have a look at the methods developed by Weatherhead et al., Factors affecting the detection of trends: Statistical considerations and applications to environmental data, J. Geophys. Res. 103, 17149-17161, 1998, doi:10.1029/98JD00995. This has been the standard method for establishing trends in atmospheric components for years. Apply the method to your results as much as possible. At least, add some discussion on the significance of the trend you found based on the well-established Weatherhead et al. method.

**Minor comments:**

- p. 2, l. 34: "SMO" has only been defined in the abstract so far, which is not a good place for an acronym definition. Please re-define here at the first use in the main text.

- p. 3, l. 16: "... signal to noise (SNR)." → "... signal-to-noise ratio (SNR)."

- p. 3, l. 27: "... contains an extra weak H2O absorption line ..."? Do you mean "extra weak" as in "very weak" or as in "an additional weak line"?

- p. 4, l. 25: what is the typical tropopause height at Maïdo? Is the 20 km range a fixed value or baiscally defined by the tropopshere height?

- p. 5, Eqns. 2 & 3: do nut use "$retrieval\ parameter\ error$" in an equation. Give it a proper mathematical symbol like $\epsilon_r$ or similar and provide a definition ("The retrieval parameter error $\epsilon_r$ is defined as ..."). Then use the symbol in your equations.

- p. 6, l. 2: why did you chose 5%? Why not more or less? Where does your information on the error distribution of the $SF_6$ profile come from?

- p. 6, l. 8: Do not use "$retrieval\ parameter\ error$" in italics. Use symbol or spell out in the same typeface as the rest of the text.

- p. 6, l. 13-21: most of the parameters and acronyms used here (zshift, ILS, Pseudo database) are defined somewhere around Sec. 2.1, about 3 pages further up in the manuscript. Could you please provide these definitions closer to the point in the manuscript where they are actually used for the first time?

- p. 8, l. 20: "... is about 0.4 years greater than ..." → "... is about 0.4 years higher than ..."

- p. 8, l. 24: "... has much more data points ..." → "... has many more data points ..."

- p. 8, l. 28: "... decreasing above tropopause, ..." → "... decreasing above the tropopause, ..."

- p. 10, l. 2: "... is not public available yet." → "... are nor publicly available yet."

- p. 10, l. 5: "... are public available ftp://..." → "... are publicly available at ftp://..."

- p. 16, Fig. 2: please add a close-up view of the $SF_6$ line as it is not really visible in the spectral overview.

---

## Author Comment (AC1) · 4 Jan 2018

*Black: referee's comments red: authors' answers*
*First, we want to thank the referee for the detailed analysis of our paper.*
*For the details, please look into the paper with keeping track of changes.*

The manuscript submitted by Zhou et al. presents measurements and trends of SF6 in the atmosphere using remote sensing techniques with ground-based FTIR spectrometers on Reunion Island. These are supplemented with analysis of measurements from in-situ instruments and space-borne sensors. This work is very significant and I recommend its publication after a few items below have been addressed/clarified.

General Comments
Column averaged dry mole fractions of SF6: To readers not familiar with this method, elaboration of this procedure is necessary. How are the dry air total columns measured/obtained? Since the XSF6 trend would depend on this, it would be important to establish and/or show the robustness of the dry air total column measurements.
We added the definition of XSF6 in the text. The uncertainty of the total column of dry air is discussed in the text, which is less than 0.1%.

Pseudo-lines: Please explain/justify the use of pseudo-lines, e.g. how are they derived? Do ACE and MIPAS use pseudo-lines as well?
Added in the text.
Pseudo-lines were produced by G.C. Toon (NASA-JPL) by fitting all the laboratory spectra simultaneously, which include mean intensities and effective lower state energies on a 0.005 cm$^{-1}$ frequency grid. These artificial lines at arbitrary positions do not represent transitions of molecules. Instead, their line-widths and intensities are fitted to the laboratory spectra such that the pseudo-line list allows to simulate the measured spectra. We use the Pseudo-lines for the FTIR retrievals, because it allows us to get a better fitting.

ACE uses HITRAN 2004 (http://www.ace.uwaterloo.ca/ACE-FTS_v2.2/ACE-SOC-0011-1D-ACE-FTS_ascii_fileformat_for_v2.2.pdf)
MIPAS uses HITRAN2000 (Haenel et al., 2015).

Smoothing error: From my understanding, the smoothing error can be evaluated if there is an independent measurement of the true profile, an estimate of its variability and covariance matrix (e.g. Barrett et al., 2002 did this for Ozone, http://onlinelibrary.wiley.com/doi/10.1029/2001JD001298/abstract). Though I understand that following this would be difficult for SF6 at the site, would the authors please comment on this?
As there is no real measurement available (e.g. balloon sondes). We apply the following method to get the systematic and random uncertainties of SF6.

Systematic uncertainty:
The SF6 is constantly increasing during last two decades, with the annual growth of ~3.0%. As the SF6 a priori profile is fixed and scaled to the in-situ measurement in the year of 2009, we assumed that there is no systematic error for 2009, but 3% for 2010 and 2008; 6% for 2011 and 2007; 9% for 2012; 12% for 2013; 15% for 2014 and 2004; 18% for 2016. After taking the number of measurements as the weighing function, the mean value is about 5%. Therefore, we apply 5% as the systematic uncertainty of SF6.

Random uncertainty:
We calculated the covariance matrix from the MIPAS measurements between 2002 and 2012 at Reunion Island. The standard deviation is about 30%, therefore we apply this value as the random uncertainty of SF6.

On the use of the retrieval window and improved fits: What criteria is used to determine a "better fit"? For example, in Fig. 2. (right panel), the residuals at 948.0 cm^{-1} seem to indicate that something is not fitted well.

The root mean square (RMS) of the fitting residual is applied to be the key parameter to check whether the fitting is better or not. The RMS is the root mean square of the difference spectrum (obs-fit).

In Fig.2 (right panel), the RMS is 0.0015, reprinting the noise level, while the intensity of the SF6 absorption line is 0.03. Such a fitting (0.0015/0.03 = 5%) is close to the retrieval random uncertainty (6.7%; see Table 2), therefore, we think the fitting is good enough to retrieve SF6.

Specific comments
Instrument and ILS: A mention of the spectral resolution used in the actual retrieval and the goodness of the ILS would be helpful to readers. For example, in Section 2.1., it was mentioned that the max OPD of the instrument is 250 cm, is this the same OPD used for SF6 measurements at both sites?
Added in the text.
The spectra of 700-1400 cm−1 at St Denis and Maïdo are recorded with the same settings. Two maximum optical path difference (MOPD) of 82 and 125 cm are operated to gather the interferogram of the direct solar radiation, and then the interferogram is transformed to a spectrum with the spectral resolution of 0.010975 and 0.0072 cm−1 through a fast Fourier transform (FFT) algorithm.

Page 6 line 10 "balloon measurements" appears here for the first time. I think this should be defined earlier in the manuscript. I assume these are sondes, right?
Yes, they are balloon sondes.

Table 2, Please elaborate on how the systematic and random errors under "SF6 Spectroscopy" were calculated.
Added in the text.

Technical Comments
Page 2, line 2: I suggest to change "SF6 emissions reduced" to "SF6 emissions decreased" Page 2, line 17: "aboard a shuttle": please specify which one, if possible.
Page 4, line 19: "In order to extract as much as possible information" -> "In order to extract as much information as possible"
Page 4, line 28: Consider rephrasing this sentence: "That means when we use the FTIR retrievals, only the total column should be trusted instead of the profile."
All corrected.

References:

Haenel, F. J., Stiller, G. P., von Clarmann, T., Funke, B., Eckert, E., Glatthor, N., Grabowski, U., Kellmann, S., Kiefer, M., Linden, A., and Reddmann, T.: Reassessment of MIPAS age of air trends and variability, Atmos. Chem. Phys., 15, 13 161–13 176, doi:10.5194/acp-15- 13161-2015, 2015.

---

## Author Comment (AC2) · 4 Jan 2018

*Black: referee's comments* *red: authors' answers*
*First, we want to thank the referee for the detailed analysis of our paper.*
*For the details, please look into the paper with keeping track of changes.*

The manuscript "Ground-based FTIR retrievals of SF6 at Réunion Island" by Minqiang Zhou et al. describes a 12-year time series of SF6 column measurements in the troposphere and lowermost stratosphere. Given the scarce observation pool of SF6 in general and its significance as a purely anthropogenic extremely long-lived greenhouse gas with a huge global warming potential, this is a very valuable data set. The authors derive an SF6 trend from their own observations as well as a comparison with two satellite datasets and one set of near-surface in-situ observations in the tropics.

The manuscript is well written and describes the data sets and the retrieval parameters and error budgets for the SF6 time series that were used in the study. To calculate trends from each dataset, a linear model with periodic (seasonal) components is applied. The resulting trend estimates are close but not identical. This is explained by different vertical and latitudinal coverage of the used datasets.

General comments:
1. One criticism that I have is the misuse of tense throughout the manuscript. Practically all of the text is written in present tense. However, the convention for scientific writing is that past tense should be used for reporting the authors' observations and results while present tense is reserved for well-known facts and cited results from the scientific literature. Please refer to guidelines on the internet such as
https://www.nature.com/scitable/topicpage/effective-writing-13815989.
Corrected.

2. I also think that there should be a map that shows the locations of the ground based observations as well as the latitude bands covered by the satellites.
Added (see Figure 4).

3. Given the fact that the SF6 spectral line is weak and the retrieval depends on the SMO observations for prior information, the significance of the derived trend(s) should be better scrutinized. Please have a look at the methods developed by Weatherhead et al., Factors affecting the detection of trends: Statistical considerations and applications to environmental data, J. Geophys. Res. 103, 17149- 17161, 1998, doi:10.1029/98JD00995. This has been the standard method for establishing trends in atmospheric components for years. Apply the method to your results as much as possible. At least, add some discussion on the significance of the trend you found based on the well-established Weatherhead et al. method.
The trend methodology in this study is same as the Basic Statistical Modeling in Weatherhead et al, 1998. I added this reference in the text.

Minor comments:
• p. 2, l. 34: "SMO" has only been defined in the abstract so far, which is not a good place for an acronym definition. Please re-define here at the first use in the main text.
Corrected.

• p. 3, l. 16: ". . . signal to noise (SNR)." → ". . . signal-to-noise ratio (SNR)."
Corrected.

• p. 3, l. 27: "... contains an extra weak H2O absorption line ..."? Do you mean "extra weak" as in "very weak" or as in "an additional weak line"?
I mean "an additional weak line", corrected.

• p. 4, l. 25: what is the typical tropopause height at Maïdo? Is the 20 km range a fixed value or basically defined by the tropopshere height?
The typical tropopause height at Maïdo is 16-17 km. Added in the text.
20 km range is a fixed value from the averaging kernel of the FTIR SF6 retrieval.

• p. 5, Eqns. 2 & 3: do not use "retrieval parameter error" in an equation. Give it a proper mathematical symbol like $\varepsilon_r$ or similar and provide a definition ("The retrieval parameter error $\varepsilon_r$ is defined as . . . "). Then use the symbol in your equations.
Corrected.

• p. 6, l. 2: why did you chose 5%? Why not more or less? Where does your information on the error distribution of the SF6 profile come from?
The SF6 is constantly increasing during last two decades, with the annual growth of ~3.0%. As the SF6 a priori profile is fixed and scaled to the in-situ measurement in the year of 2009, we assumed that there is no systematic error for 2009, but 3% for 2010 and 2008; 6% for 2011 and 2007; 9% for 2012; 12% for 2013; 15% for 2014 and 2004; 18% for 2016. After taking the number of measurements as the weighing function, the mean value is about 5%. Therefore, we apply 5% as the systematic uncertainty of SF6.

• p. 6, l. 8: Do not use "retrieval parameter error" in italics. Use symbol or spell out in the same typeface as the rest of the text.
Corrected.
• p. 6, l. 13-21: most of the parameters and acronyms used here (zshift, ILS, Pseudo database) are defined somewhere around Sec. 2.1, about 3 pages further up in the manuscript. Could you please provide these definitions closer to the point in the manuscript where they are actually used for the first time?
Corrected.
• p. 8, l. 20: "... is about 0.4 years greater than . . . " → "... is about 0.4 years higher than . . . "
Corrected.
• p. 8, l. 24: "... has much more data points . . . " → "... has many more data points . . . "
Corrected.
• p. 8, l. 28: "... decreasing above tropopause, . . . " → "... decreasing above the tropopause, . . . "
Corrected.
• p. 10, l. 2: "... is not public available yet." → "... are not publicly available yet."
Corrected.
• p. 10, l. 5: "... are public available ftp://. . . " → "... are publicly available at ftp://. . . "
Corrected.
• p. 16, Fig. 2: please add a close-up view of the SF6 line as it is not really visible in the spectral overview.
Added.